# Impact of Molecule Concentration, Diffusion Rates and Surface Passivation on Single-Molecule Fluorescence Studies in Solution

**DOI:** 10.3390/biom12030468

**Published:** 2022-03-18

**Authors:** Olessya Yukhnovets, Henning Höfig, Nuno Bustorff, Alexandros Katranidis, Jörg Fitter

**Affiliations:** 1AG Biophysik, I. Physikalisches Institut (IA), RWTH Aachen University, 52074 Aachen, Germany; h.hoefig@fz-juelich.de; 2Ernst Ruska-Centre for Microscopy and Spectroscopy with Electrons (ER-C-3), Institute of Biological Information Processing IBI-6, Forschungszentrum Jülich, 52425 Jülich, Germany; n.bustorff@fz-juelich.de (N.B.); a.katranidis@fz-juelich.de (A.K.)

**Keywords:** single-molecule Förster resonance energy transfer, burst analysis, two-color coincidence detection, confocal fluorescence microscopy, chance coincidence probability

## Abstract

For single-molecule studies in solution, very small concentrations of dye-labelled molecules are employed in order to achieve single-molecule sensitivity. In typical studies with confocal microscopes, often concentrations in the pico-molar regime are required. For various applications that make use of single-molecule Förster resonance energy transfer (smFRET) or two-color coincidence detection (TCCD), the molecule concentration must be set explicitly to targeted values and furthermore needs to be stable over a period of several hours. As a consequence, specific demands must be imposed on the surface passivation of the cover slides during the measurements. The aim of having only one molecule in the detection volume at the time is not only affected by the absolute molecule concentration, but also by the rate of diffusion. Therefore, we discuss approaches to control and to measure absolute molecule concentrations. Furthermore, we introduce an approach to calculate the probability of chance coincidence events and demonstrate that measurements with challenging smFRET samples require a strict limit of maximal sample concentrations in order to produce meaningful results.

## 1. Introduction

One major approach to investigate biomolecule conformations, dynamics, or molecular interactions on single molecule level is given by fluorescence-based techniques. In this respect, single-molecule Förster resonance energy transfer (smFRET) is one of the most powerful techniques to study the structural dynamics of biological macromolecules, such as proteins or protein complexes (for details, see [1,2]). Many smFRET applications make use of confocal microscopy, for which a diffraction limited small detection volume is created by an optical setup that allows for the measurement of fluorescence emissions from individual diffusing molecules during a rather short time period of the molecule’s transit through this detection volume [3,4]. For medium-sized proteins and confocal detection volumes in the order of a few femtoliters, the dwell time of the diffusing molecule is in the millisecond time regime. During this time, the emitted “burst of photons” from individual molecules represents the essential feature of a single molecule [4,5]. With the help of powerful burst analysis tools, the relevant data can be extracted from typical time trace measurements [6,7]. In this way, up to a few thousand bursts can be measured within hours, delivering reasonable counting statistics for a proper single-molecule analysis [4,8]. 

An important prerequisite to achieving single-molecule sensitivity by using this approach is that the absolute molecule concentration has to be kept to a value that corresponds to the presence of not more than one molecule (or one type of molecule) at the time in the confocal detection volume. Due to the stochastic character of the Brownian diffusion the molecules perform in solution, it is typically assumed that this requirement is fulfilled if the average number of molecules in the detection volume (around a few femtoliters) is given by <*N*> ≈ 10^−2^; see, for example, [9]. In most applications with diffraction limited detection volumes in confocal microscopy, this requirement is related to molecule concentrations of a few tens of pico-molars. In addition to smFRET, several further fluorescence-based approaches are employed to study the assembly and disassembly of macromolecular complexes, antigen–antibody interactions, or DNA hybridization. For example, two spectrally different fluorophores (attached to corresponding biomolecular binding partners) can be excited simultaneously, and forms of two-color coincidence detection (TCCD) can be used to measure the binding of the two interaction partners [10,11,12,13,14]. The advantage of this approach is related to the fact that molecules down to the pico-to femtomolar concentration range can be analyzed, thus offering an excellent method to probe molecular interaction with extreme high affinities (with dissociation constants in the picomolar regime). For TCCD studies [10,14], one need not only measure at rather low molecule concentrations (in order to avoid random or chance coincidence). In addition, the absolute values of concentrations of the two differently labelled species also need to be known as precisely as possible, for example, in order to calculate reliable dissociation constants. Furthermore, the average number of molecules in the detection volume at the time depends not only on the molecule concentration but also on the diffusion rate of the molecules. 

Here, we discuss specific single-molecule Förster resonance energy transfer (smFRET) and TCCD applications for which high demands on sample properties are required in order to obtain reliable results. In this respect, methodical approaches are introduced that help to monitor and control the molecule concentration. Finally, a reduced quality of the obtained results is demonstrated for data that are measured with samples at a sub-optimal concentration regime. 

## 2. Materials and Methods

### 2.1. Double-Stranded DNA

The hybridization procedure and detailed sample preparation procedures were published previously [15]. Briefly, dsDNA samples were prepared by hybridizing a labelled ssDNA strand 5′-GGA CTA GTC TAG GCG AAC GTT TAA GGC GAT CTC TGT TTA CAA CTC CGA-3′ with an unlabelled ssDNA strand 5′-TCG GAG TTG TAA ACA GAG ATC GCC TTA AAC GXT CGC CTA GAC TAG TCC-3′ (IBA, Göttingen, Germany). The labelled strand either had an Alexa488 dye at the 5′ end and an Atto647N dye at the 3′ end or just a single Atto647N dye at 3′ end to produce a double- or a single-labelled dsDNA, respectively. After hybridization, samples were aliquoted and stored at −20 °C. If not stated otherwise, the dsDNA samples were measured in a DNA standard buffer: 20 mM Tris (pH 7.5, Carl Roth, Karlsruhe, Germany), 100 mM NaCl (Carl Roth, Karlsruhe, Germany), and 10 mM MgCl_2_ (VWR, Langfeld, Germany). 

### 2.2. Phosphoglycerate Kinase

Phosphoglycerate kinase (PGK) expression procedure is described in detail in [16]. Protein labelling and purification is described in [17]. Briefly, a single cysteine mutant PGK C97S Q135C was produced by site-directed mutagenesis. Protein samples were frozen in liquid nitrogen and stored at −80 °C (New Brunswick Scientific GmbH, Nürtingen, Germany). The single cysteine mutant was labelled either with maleimide-functionalized Alexa647 or Alexa488 by incubating 10 μM PGK solution with fivefold excess of dye. After labelling, the protein was purified from unbound dye excess by size exclusion chromatography, using a Sephadex G25 packed column (GE Healthcare Bio-Sciences, Uppsala, Sweden). Labelled protein samples were stored at 4 °C for a maximum of a few days.

### 2.3. Ribosomes

The process of isolation and labelling of ribosomes is described in [14]. Ribosomes were isolated by zonal centrifugation and incubated with 20-fold excess of Cy5-NHS-ester-functionalized dye (GE Healthcare Life Sciences, Little Chalfont, United Kingdom). Labeled ribosomes were purified by pelleting the ribosomes through a 1.1 M sucrose cushion in Tico buffer: 20 mM HEPES (pH 7.6, Carl Roth, Karlsruhe, Germany), 10 mM magnesium acetate (Ac_2_Mg, Carl Roth, Karlsruhe, Germany), 30 mM ammonium acetate (AcNH_4_ Carl Roth, Karlsruhe, Germany), and 4 mM β-mercaptoethanol (β-ME) (Carl Roth, Karlsruhe, Germany). The concentration of Cy5 and ribosomes was determined spectroscopically in a NanoDrop spectrophotometer (Thermo Fisher Scientific, Waltham, MA, USA) using the absorption coefficients ε_cy5_ = 2.5 × 10^5^ M^−1^ cm^−1^ (at λ = 647 nm) and ε_70S_ = 4.2 × 10^7^ M^−1^ cm^−1^ (at λ = 254 nm), respectively. The label ratio was calculated to be ≈6 Cy5 dyes per 70S ribosome.

### 2.4. Ribosome Nascent Chain of Calmodulin

Ribosome nascent chains (RNCs) of calmodulin (CaM) were synthesized to full polypeptide chain length (149 amino acid residues) using a cell-free protein synthesis system [18]. In order to keep the nascent chain bound to the ribosome, an enhanced arrest peptide sequence from *E. coli* protein SecM, named SecMstr (FSTPVWIWWWPRIRGPP), was used [19]. The arrest peptide was introduced downstream of CaM connected via a linker composed of a 30 amino acid long sequence of glycines and serines spanning the length of the ribosomal tunnel. At the N-terminus, a twin strepII tag (WSHPQFEKGGGSGGGSGGSAWSHPQFEK) was also introduced for affinity purification. Two unique unnatural amino acids (UAAs), namely, CpK and AzF were incorporated co-translationally at determined positions (34 and 110). The side chains of these two UAAs have functional groups that can react with functionalized dyes to produce double-labelled RNCs, with an inter-dye distance suitable to observe FRET [20]. The reactions were performed using an *E. coli* high yield (Biotechrabbit, Berlin, Germany) or a PURExpress (NEBiolabs, Ipswich, MA, USA) cell-free system. The reaction was stopped by adding two volumes of RNC buffer (20 mM HEPES (pH 7.5), 50 mM Ac_2_Mg, 30 mM AcNH₄, 0.5 mM TCEP, 0.005% Tween20) (Sigma-Aldrich GmbH, Steinheim, Germany). RNCs were purified using magnetic StrepTactinXT (IBA Lifesciences GmbH, Göttingen, Germany) beads and labelled sequentially with 50 μM of red AF647-DBCO and blue AF488-tetrazine (Click Chemistry Tools, Scottsdale, AZ, USA) dyes. After each labelling step, the excess of dye was removed using Zeba columns (Thermo Fisher Scientific, Waltham, MA, USA), previously equilibrated with apo-buffer (10 mM EGTA, 50 mM MOPS, 150 mM KCl, 0.005% Tween20) (Sigma-Aldrich GmbH, Steinheim, Germany) [21], for subsequent smFRET measurements.

### 2.5. Confocal Microscopy

Confocal measurements were performed using a MicroTime200 (PicoQuant, Berlin, Germany). The fluorophores were excited using LDH-D-C 485B and LDH-D-C 640B lasers with 485 nm and 640 nm emission (PicoQuant, Berlin, Germany) and a power of 21 μW and 18 μW, respectively. For smFRET and Brightness-gated two-color coincidence detection (BTCCD) measurements, lasers were operated in a pulsed-interleaved excitation (PIE) scheme, in which blue and red excitation was alternated in order to directly excite both channels [22]. The excitation light was focused and collected by a high numerical aperture water immersion objective (UPLSAPO 60×; Olympus, Hamburg, Germany) and directed through a 75 μm pinhole. The emission signal was separated by a dichroic mirror (T600lpxr, Chroma Technology, Olching, Germany) and filtered by band pass filters of 535 nm (FF01-535/55-25, Semrock, Rochester, NY, USA) and 685 nm (ET685/80m, Chroma Technology, Olching, Germany) for the blue and red channels, respectively. Photons were detected by single-photon avalanche diodes (τ-SPAD, PicoQuant, Berlin, Germany; COUNT-T, Laser Components, Olching, Germany).

### 2.6. Sample Preparation and Data Acquisition

Unless stated otherwise, all samples were measured on PEGylated cover slides. For the PEGylation procedure, high-precision cover glasses of 170 μm thickness (Marienfeld, Lauda-Königshofen, Germany) were cleaned with Piranha solution, plasma cleaned, treated with silane, and left to react overnight with NHS-functionalized PEG (MW = 750 Da, Rapp Polymere, Tübungen, Germany) solution. Concentration of all samples was first determined with FCS, and then samples were diluted to aimed single-molecule concentrations. Afterwards, the real single-molecule concentration was determined with bursts analysis. Single-molecule sample aliquots were measured within 10–20 min time intervals that were summed up afterwards for the analysis. The typical data acquisition time was about 60–180 min for a dataset. In order to maintain constant concentration and avoid evaporation during a long measurement time period, sample holders were sealed with parafilm.

### 2.7. Burst Analysis

The inter photon lag (IPL) trace was calculated for acquired intensity traces [23]. Single bursts in both the red and the blue channels were discriminated from the background by applying a suitable threshold (usually ≈100 μs). Bursts that contained only one photon were discarded because they would induce an artificially small dwell time and low molecular brightness. The start time of each burst corresponded to the macro time tag of the first photon of that burst, and, accordingly, the end time of a burst was defined as the macro time tag of the last photon. The burst duration was defined as the difference between its start and end time [6,7,8]. Bursts with a duration time that was more than 100-fold longer than the average burst duration time and bursts with the number of photons of more than 100-fold more than the mean number of photons per burst were considered as aggregates or contamination and were removed from the analysis. Typical datasets contained 10^3^–10^4^ accepted bursts.

### 2.8. Brightness-Gated Two-Color Coincidence Detection

Brightness-gated two-color coincidence detection (BTCCD) is a method realized by means of simultaneous single-molecule two-color confocal detection to quantify the fraction of bound (coincident) molecules. In contrast to conventional two-color coincidence detection (TCCD; see, for example, [10]), BTCCD overcomes the problem of coincidence fraction underestimation, caused by incomplete detection volume overlap for different excitation wavelengths and lens aberrations. In order to precisely estimate the coincidence fraction, only molecules that diffused through both confocal volumes should be considered for analysis. In practice, it is assumed that the corresponding molecule trajectories resulting in bright bursts with a high number of photons are more likely to touch both volumes, whereas dim bursts with only a small number of photons are more likely to only slightly touch one of the volumes. For each accepted burst, the burst intensity, i.e., number of photons detected between the start and end time, and the mean number of photons per burst was calculated. To perform a coincidence analysis, the brightness threshold *n_br_*, defined as the number of photons in a burst, normalized to the mean number of photons, was continuously increased. The coincidence was calculated for the red channel (*f_RB_*) and the blue channel (*f_BR_*) independently with
(1)fRB(nbr)=NRB(nbr)NR(nbr), fBR(nbr)=NBR(nbr)NB(nbr)
where *N_RB_* and *N_BR_* are the number of coincident bursts in the red and blue channels, respectively, and *N_R_* and *N_B_* are the total number of selected red and blue bursts, respectively. For each value of the brightness threshold, only bursts that had more photons as defined by the brightness threshold were considered for analysis. Two bursts were considered as coincident if the start or end time tag of one burst was within the start and end time tags of the other burst. Coincidence fraction increased with the increase of *n_br_* and eventually saturated once all bursts considered for the analysis corresponded to molecule trajectories though both volumes. A more detailed description of the BTCCD method can be found in the Appendix A, and in [14].

### 2.9. Single-Molecule FRET Analysis

After selecting bursts by means of burst analysis, photon counts were calculated for each accepted burst. The use of the PIE excitation scheme allowed us to calculate photon counts for acceptor emission after donor excitation (*I_AD_*), acceptor emission after acceptor excitation (*I_AA_*), and donor emission after donor excitation (*I_DD_*), as well as to eliminate donor-only and acceptor-only populations from the analysis. For bursts with *I_AD_* + *I_DD_* > 20 photon counts, the energy transfer efficiency
(2)E=IADIAD+γ⋅IDD
was calculated, where *γ* is a correction factor accounting for differences in dye quantum yields and detection efficiencies in donor and acceptor channels. The stoichiometry parameter is given by
(3)S=IAD+IDDIAD+IDD+IAA

Furthermore, *E* and *S* values were corrected for crosstalk and direct excitation as described in [24]. In order to evaluate the quality of the obtained smFRET data, FRET efficiency histograms and two-dimensional efficiency–stoichiometry histograms were plotted. Finally, FRET efficiency histograms were fitted with Gaussians in order to characterize individual subpopulations.

### 2.10. Concentration Determination in Single-Molecule Measurement

The most common approach to determine the molecule concentration in confocal spectroscopy is to make use of <*N*> = 1/G(τ = 0) in FCS analysis. However, remaining sources of background are typically not correlated and affect the correlation function only by lowering its amplitude *G*(0), thus leading to overestimation of the number of the fluorescent particles of interest [25]. The effect is especially prominent in samples containing very low fluorophore concentrations, and also explicit consideration of background counts cannot exhibit reliable values of <*N*> for concentrations below ≈100 pM. Therefore, we made use of an alternative approach based on burst analysis. In order to calculate the molar concentration of a sample *C = N_av_ /(N_A_∙V_eff_)* in a single-molecule experiment, the average number of detected molecules *N_av_* (for simplicity, from now on depicted only by *N*) and the dimension of confocal detection volume Veff=π3/2ω02z0 need to be known (*N_A_*: Avogadro constant). The latter is directly accessible from FCS calibration measurements. *N* can be calculated from the total number of detected bursts, *B_meas_*, and the duration time of detected bursts, τ_d_, by considering the total fluorescence time *t_F_* [4]. Total fluorescence time was defined as the product of total measurement time *t_meas_* and the probability to detect a molecule (1 − exp(−*N*)) or by the product of *B_meas_* and τ_d_
(4)tF=1−exp−N⋅tmeas=Bmeas⋅τd

This equation can now be used to determine the average number of detected molecules:(5)N=−ln1−Bmeasτdtmeas

Strictly, the determination of *N* is fully correct only for a single species for which we can provide a single (averaged) τ*_d_* value. 

Typical values for calculated molar concentrations as obtained from confocal measurements with different pinhole diameters are shown in Table 1.

### 2.11. Chance Coincidence Probability

In order to perform a single-molecule experiment, the probability to detect more than one molecule must be negligibly low [26]. Knowing the probability of the multi-molecule event detection (i.e., having a number of current molecules *N_c_* > 1 in the detection volume) allows for the maintenance of a proper single-molecule concentration regime [27]. As shown in [28], it is calculated from Poisson distribution, with the probability to detect two molecules at a time being given by
(6)pmn=2(N)=exp−2N1−exp−N

The probability to detect more than two molecules (*N_c_* > 2) is negligibly small for *N* << 1. Therefore, we define chance coincidence events as the presence of two molecules at a time in the detection volume. 

For BTCCD, chance coincidence only matters for cross-color multi-molecule events, i.e., when a blue-labelled molecule enters the volume while the red-labelled molecule is in the volume and vice versa. The probability of detecting a blue-labelled molecule during the dwell time of a red-labelled molecule will depend on burst duration time in red and blue channels, τdR and τdB, respectively, and on the average number of molecules in the blue channel, *N_B_.*
(7)pRB=1−exp−τdRτdBNB

Blue-labelled molecules can either already be in the detection volume or enter during the dwell time of a red-labelled molecule, meaning that both probabilities need to be added. Moreover, because only cross-color events should be considered, only red-only molecules will cause a chance coincidence detection, and thus a (1 − *f_RB_*)-factor should be introduced. The chance coincidence fraction is then calculated as
(8)fRBChance=1−fRB1−exp−NBτdRτdB+1

In the same manner, the chance coincidence fraction accounting for detecting a red-labelled molecule during the dwell time of a blue-labeled molecule is
(9)fBRChance=1−fBR1−exp−NRτdBτdR+1

For the calculation of theoretical chance coincidence fractions, we made use of experimentally determined *N*- and τ*_d_* values (see Section 2.10). A more detailed derivation of the presented relations is given in ref. [29] and in the Appendix A.

## 3. Results and Discussion

Practice samples for confocal fluorescence spectroscopy often consist of a rather heterogeneous ensemble of labelled molecules. This heterogeneity can originate from the fact that (i) the biological molecules of interest are present in different conformational states, or that (ii) individual biomolecules exhibit different degrees of labelling. For example, in the case of FRET studies that require donor and acceptor double-labelled species, donor-only or acceptor-only species might be present in the sample solution. In most single-molecule studies, problems related to such sample labelling heterogeneity can either be avoided by highly productive sample preparation and purification procedures [30,31] or circumvented by elaborated molecule sorting algorithms [22,24]. However, in some cutting-edge applications, sample heterogeneity can still cause serious problems. In this respect, we focus here mainly on two types of samples: (i) Proteins that are produced by cell-free synthesis systems, allowing for the incorporation of unnatural amino acids for more selective double dye conjugation. Although these smFRET samples offer the opportunity for studying co-translational protein folding or multi-protein complex assembly, the obtained proteins often suffer from rather incomplete labelling and low protein yields [20,32]. As a consequence, the typically applied protein purification procedures (for example, to remove free dyes) also work less efficiently. (ii) The strength of the BTCCD approach is to determine bi-molecular binding affinities in the low pico-molar regime. This also requires low molecule concentrations of the individual complementary single-labelled binding partners. For both mentioned types of samples, a rather low molecule concentration in combination with a certain tendency for unspecific molecule attachment to the cover-slide surface may cause problems and artifacts, which is discussed in the following subchapters. 

### 3.1. Unbalanced Reduction of Molecule Concentrations during Extended Measurement Times

In typical BTCCD studies, calibration measurements have to be performed in order to validate the fact that brightness-gating (the use of increasing *n_br_* values) reasonably reduces the effects of the confocal detection volume mismatch (related to the two different excitation wavelengths) [14]. For this purpose, a double-stranded DNA molecule labelled with two different dyes was employed. Since the label ratio for each color (i.e., each dye with its respective absorption wavelength) was very high, typically larger than 95%, we obtained very high coincidence fractions *f_RB_* and *f_BR_* (see Section 2.8, Equation (1)). In the case of dsDNA (length 48 bp) labelled with Alexa488 and Atto647N, the expected high coincidence fractions were visible for *n_br_* > 2 (see Figure 1a). A closer look at Figure 1a demonstrates that the coincidence fraction *f_RB_* (blue) decreased substantially over a time span of 60 min, while the coincidence fraction f*_BR_* (red) remained more or less unchanged during the same time span. This result is shown more clearly in Figure 1c, wherein *f_RB_* dropped from 87% to 68%, and *f_BR_* exhibited a constant value of about 90%. Such behavior can be explained by the following facts: (i) There was a small fraction of single-labelled species, 3–5% only labelled with Alexa488 or with Atto647N. (ii) There is always a certain, often rather small, fraction of molecules that nonspecifically sticks to the surface, even in the case of PEGylated cover-slides. In our case, one fluorescent dye seemed to show a tendency for stickiness, and clearly it was the Atto647N that exhibited this tendency. As a consequence, the total molecule concentration of dsDNA molecules dropped down over time. Since the majority of the diffusing molecules was still double labelled, the concentration for both colors was reduced from approximately 18–20 pM to 6–8 pM after 60 min (see Figure 1b). However, a further consequence was that the single-labelled Atto647N molecules were systematically extracted from the solution, while at the same time, the single-labelled Alexa488 molecules remained in the solution. Exactly this mismatch caused a decreasing *f_RB_* (blue) and a constant *f_BR_* (red) over time.

By adding a small amount of the detergent Tween20 to the buffer solution, the stickiness of Atto647N almost vanished (see Figure 2a,b). The corresponding results from measurements with Tween20 exhibited nearly constant molecule concentrations for all species, as well as stable and almost identical coincidence fractions *f_RB_* and *f_BR_* (for *n_br_* values above a certain threshold), as expected for this type of sample (see [14]). 

In another case, we re-evaluated smFRET data from measurements performed in a multidomain protein folding study [17]. In this study, the conformational structure of phosphoglycerate kinase (PGK) was measured as a function of the chemical denaturant GuHCl. As observed already in various studies, proteins typically undergo a structural expansion upon increasing GuHCl concentrations, representing an unfolding transition [5,33]. The corresponding succession of smFRET histograms first showed a population for the native state (compact high FRET state). With increasing GuHCl concentrations, a further population of the unfolded state showed up, while well above the GuHCl half-transition concentration (for PGK C_1/2_ ≈ 0.65 M), only the peak of the unfolded state (expanded low FRET state) remained. Since GuHCl is not only an efficient chemical denaturant, but also a good solvent (for molecules that expose hydrophobic regions), the GuHCl concentration determines not only the degree of unfolding, but also the tendency of partly unfolded proteins to stick to the cover-slides during extended measurement times. 

In this respect, we observed for proteins in buffers with GuHCl concentrations close to C_1/2_ drastically reduced count rates during confocal smFRET measurements. A more detailed inspection of the data measured at C_1/2_ revealed that the number of detected bursts and thereby the molecule concentration was reduced, mainly caused by unspecific protein binding to the cover-slide surface (see Figure 3a). Interestingly, we did not observe a similar count rate reduction in other GuHCl concentration regimes (for example, 0–0.5 M for mainly folded states and >0.9 M for mainly unfolded states) (see the Appendix A). Since at C_1/2_ we are dealing with samples that include two different subpopulations (folded and unfolded states), the question arises as to whether one of the subpopulations is more prone to surface sticking and thereby would bias the statistical weights of the obtained subpopulations. In order to answer this question, we separated the dataset that accumulated bursts over 60 min into two parts. The first part included bursts of the first 30 min (Figure 3b), and a second part that of the second 30 min (Figure 3c). This rough estimate (a more finely subdivision of the data was not possible due to the limited number of bursts per time interval) exhibited more or less identical smFRET histograms. Importantly, the statistical weights (P_1_ and P_2_) for the individual subpopulations did not vary during the total measuring time. This indicates that both protein conformations had more or less the same tendency for surface sticking. Therefore, at least in the case presented here, unspecific surface sticking of the investigated biomolecules did not bias the obtained results. 

### 3.2. Impact of Chance Coincidence on BTCCD and smFRET Results

In studies with diffusing molecules targeting single-molecule sensitivity, the molecule concentration must be low enough to ensure that we detect only single molecules. On the other hand, we would like to work with molecule concentrations as high as possible, in order to obtain a high number of bursts within a certain time interval. For achieving a reasonable trade-off between both requirements, it is worth to establishing an approach to identify optimal target concentrations, depending on the specific boundary conditions. In confocal microscopy, in principle, several freely diffusing molecules can be present in the detection volume at the same time, also known as chance coincidence. Here, we make use of mathematical terms for calculating fractions of chance coincidence, derived from some basic principles (see Section 2.11).

In order to quantify chance coincidence in a straightforward manner, we performed BTCCD analyses with samples containing a pair of two distinct molecules that do not exhibit an appreciable binding affinity to each other. The two involved molecules were labelled with different dyes, and the measured coincidence fraction can be interpreted as a pure chance coincidence. In a first example, two PGK species were labelled either with Alexa488 or with Alexa647, mixed in a certain stoichiometry and adjusted to different sample concentrations. In Figure 4a, the measured coincidence fractions *f_RB_* (solid lines) and the corresponding calculated values (dashed lines, see Equation (8)) are shown for three different molecule concentrations (here the concentration is given for the Alexa488 labelled species). 

It is obvious from this figure that only for a concentration of 30 pM was the chance coincidence fraction below 0.1, whereas for higher concentrations, *f_RB_* values were larger than 0.2 (for 57 pM) or even larger than 0.8 (for 382 pM). The obtained results indicate that only the 30 pM sample to some extent allowed for a reliable single-molecule data interpretation (with approximately 90% of all bursts originating from single molecules). In the other cases (57 and 382 pM), too many of the detected bursts contained photons from more than one molecule (20–80%). Furthermore, the presented analysis demonstrated a rather good agreement between measured and calculated *f_RB_* values. According to Equation (8) (see Section 2.11), the chance coincidence given by *f_RB_* values depended not only on the molecule concentration (given by *N*), but also on the ratio of the involved burst duration times τdR and τdB of both diffusing species. Therefore, the relative sizes of the diffusing molecules had a clear impact on the obtained chance coincidence. This led to a more pronounced chance coincidence (i.e., larger *f_RB_* values) if the red-labelled molecules diffused much slower than the blue-labelled ones. A confirmation of such a behavior is illustrated in Figure 4b, showing data obtained from a molecule pair consisting of red-labelled dsDNA and free Alexa488 (blue). In this case, even a molecule concentration of 23 pM provided more than 20% of all bursts that contained photons from more than one molecule (i.e., a single-molecule interpretation of individual burst was no longer valid). Such a behavior was even more pronounced for a sample with a very large red-labelled ribosome and free Alexa488 (see Figure 4c). The last two cases exemplify the fact that only samples with a molecule concentration of a few pM allow for a burst analysis with single-molecule interpretation. It is noteworthy that for these conditions, the required *N* value was below 0.01 (for smFRET studies, often the recommend *N* value is 0.01–0.03). Furthermore, we observed a slightly lower agreement level between calculated and measured *f_RB_* values for the samples shown in Figure 4b,c, particularly for the higher chance coincidence probability.

Finally, we analyzed smFRET data from a special case study wherein the impact of chance coincidence was clearly visible. In a challenging study to investigate protein conformations of ribosome-bound nascent chains (RNCs), smFRET data were measured for double-labelled calmodulin (CaM), which remained bound to the ribosome during the measurements. The respective samples were produced by using cell-free protein synthesis (CFPS). For a specific dye attachment of the donor and the acceptor only to the CaM protein, we made use of unnatural amino acids in order to establish a dye-binding chemistry that is orthogonal to typically used cysteine-based dye binding (see Section 2.4). Due to the rather small protein yields of CFPS, we obtained only a small fraction of double-labelled molecules (i.e., donor and acceptor). Furthermore, the sample purification methods did not work as efficiently as compared to standard applications. The smFRET results from a burst analysis (including PIE; for details, see Section 2.5 and Section 2.9) are shown for a first sample in Figure 5a. The obtained data were measured with a sample containing double-labelled molecules, incomplete labelled molecules, and free dye. For the donor color, a concentration of 30 pM was measured, whereas for the acceptor color, a concentration of 6 pM was measured. It was obvious that the corresponding transfer efficiency histogram (upper panel) did not show any reasonable FRET population that would correspond to one of the known CaM conformations (with dyes at positions 34 and 110; see, for example, [21,34]). For a second sample, the labelled molecules were much more diluted (donor color with 3 pM and acceptor color with 0.3 pM), and the corresponding smFRET histogram differed significantly from the first one (see Figure 5b). Although the number of detected bursts was still low, we were therein clearly able to identify a population centered at <E> ≈ 0.55 that represented a calcium-free state of calmodulin (apoCaM) [21,34]. This is in full agreement with the buffer conditions used in these measurements (see Section 2.4). This example demonstrated that at higher molecule concentration, the impact of chance coincidence can cause artificial FRET populations in the efficiency histogram. In particular, the peak at <E> ≈ 0 (which is often denominated “donor-only” peak but should not appear here due to the application of PIE sorting) was caused by the excess of free unbound donor dyes (which could not be removed efficiently during the sample purification).

As demonstrated in Figure 4c, this can cause a significant amount of chance coincidence with artificially too high donor emission. Bursts exhibiting the correct transfer efficiency (i.e., <E> ≈ 0.55) of the rather few “really double-labelled” molecules were hidden at these high-molecule concentrations and only became visible at much lower molecule concentrations, wherein chance coincidence vanished. Supporting our interpretation of the results shown in Figure 5, we performed further measurements with a dsDNA test sample in which only 3% of all molecules were double-labelled with donor and acceptor dyes, while the remaining dsDNA molecules were single-labelled (see the Appendix A). For this dsDNA test sample at low molecule concentration, a reasonable FRET peak was visible, but at higher concentrations (containing a considerable fraction of single-labelled species), this FRET peak disappeared. 

## 4. Conclusions

Single-molecule studies can provide valuable and unique information about the ensemble of molecules in a sample. By the identification and the separation of subpopulations, detailed knowledge about properties of individual molecules is accessible. Furthermore, the statistical weights of different subpopulations can be determined quite precisely. However, unrealized sample conditions can cause artifacts that may significantly bias the obtained results. 

Here, we demonstrated in a BTCCD analysis that it is worth determining the absolute molecule concentrations for all different species present in the sample and to monitor these values during extend measuring times. With this information at hand, unbalanced nonspecific molecule attachment to cover-slide surfaces can be identified and often prevented by slightly optimized environmental sample conditions. Furthermore, this information provides direct access to the probability of chance coincidence appearance. As shown in a smFRET study, the obtained results can suffer from chance coincidence, and only a reduction of the molecule concentration can help to overcome this problem for a given sample. Following the presented approach is particularly helpful for cutting-edge applications where the sample quality is well below that of standard applications (see, for example, [35,36]). Compared to smFRET studies, in BTCCD analyses, the situation is more favorable. We introduced an approach to calculate chance coincidence probabilities ab initio that exhibit a reasonable agreement with experimentally determined values. Therefore, we can make use of these calculations in typical BTCCD applications, for example, the determination of binding affinities based on measured coincidence fractions [14,37]. Here, the advantage is that we can tolerate chance coincidence in these kinds of measurements as long as we can accurately quantify chance coincidence fractions, because, finally, we can subtract them from the measured coincidence fractions that determine the binding affinity in the given sample. 

## Figures and Tables

**Figure 1 biomolecules-12-00468-f001:**
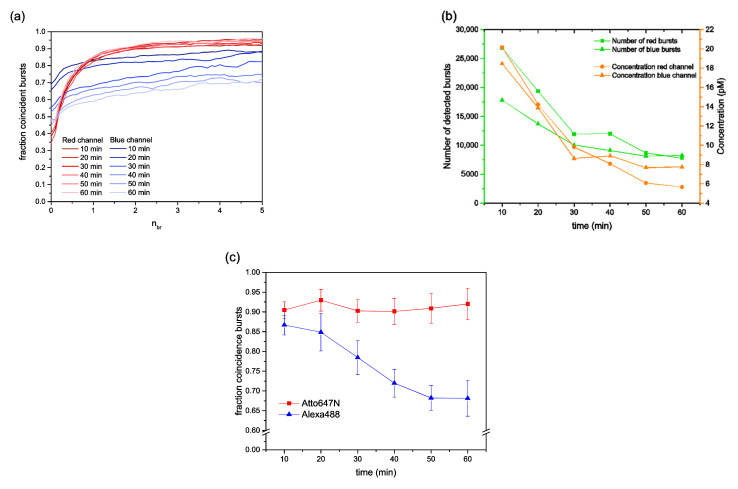
BTCCD results of double-labelled dsDNA samples as obtained from measurements at standard buffer conditions (only PEGylated surfaces, see Section 2.1 and Section 2.6). (**a**) Coincidence fractions, *f_RB_* (blue) and *f_BR_* (red), are shown as a function of the brightness-gating parameter *n_br_* for consecutive 10 min time intervals after the measurements were started. (**b**) On the basis of the data shown in (**a**), the time course of the dye concentration in solution was provided in terms of “total number of bursts” as obtained in the respective time interval (see the left y-axis) and in the molar concentration (see the right y-axis). (**c**) On the basis of the data shown in (**a**), the time course of the individual coincidence fractions *f_RB_* (blue) and *f_BR_* (red) are shown.

**Figure 2 biomolecules-12-00468-f002:**
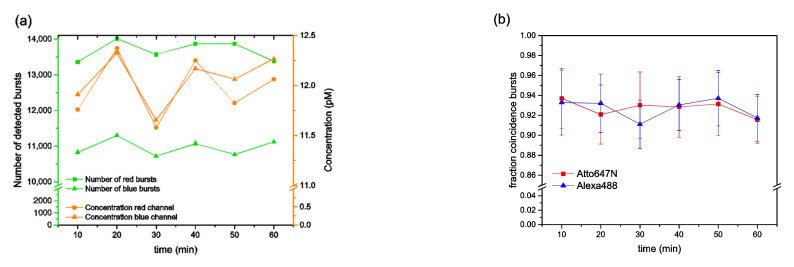
BTCCD results of double-labelled dsDNA samples. (**a**,**b**) For data which were obtained from measurements in buffers containing 0.005% Tween20, the same types of graphs are shown as in Figure 1b,c, respectively.

**Figure 3 biomolecules-12-00468-f003:**
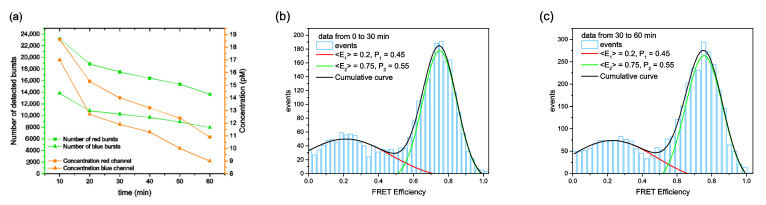
Results as obtained from smFRET measurements with PGK at 0.65 M GuHCl denaturant concentration. (**a**) The number of detected bursts and the related molar protein concentration was given during the total measuring time (60 min) for time intervals of 10 min. (**b**,**c**) The corresponding smFRET transfer efficiency histograms are shown for the first 30 min (**b**) and for the last 30 min (**c**). In both histograms, the unfolded state population (at <E_1_> ≈ 0.2) and the folded state population (at <E_2_> ≈ 0.75) were within the limits of error of the same statistical weights P_1_ ≈ 0.45 and P_2_ ≈ 0.55, respectively (area under the respective fitted Gaussian peaks).

**Figure 4 biomolecules-12-00468-f004:**
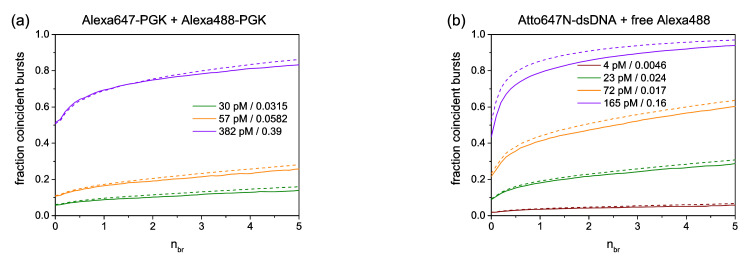
Measured (solid lines) and calculated (dashed lines) *f_RB_* values obtained from samples with a pair of two distinct individually and freely diffusing molecules, one labelled red (Alexa647/Atto647N/Cy5) and the other labelled blue (Alexa488). The respective molecule concentrations are given for the blue-labelled species (including the parameter *N*, the corresponding average number of molecules in the detection volume at the same time). The concentration of the red-labelled species (not given explicitly) was in the same regime as the blue-labelled species. The presented examples exhibited a clear variation in the relative sizes, and thereby in the corresponding burst duration times, τ_d_, of the respective molecules in the mixed samples: (**a**) similar size of red- and blue-labelled molecules, (**b**) larger red-labelled molecule versus smaller blue-labelled molecule (i.e., only Alexa488), and (**c**) very large red-labelled molecule versus small blue-labelled molecule.

**Figure 5 biomolecules-12-00468-f005:**
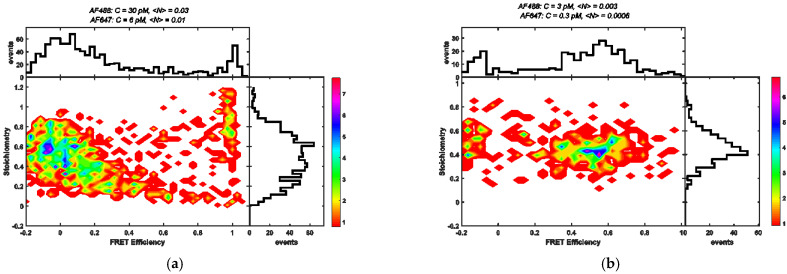
Here, stoichiometry versus FRET efficiency plots are shown for RNC samples as measured with different molecule or dye concentrations. (**a**) At higher concentrations, the corresponding FRET efficiency histogram (upper panel) showed only peaks at <E> ≈ 1 and <E> ≈ 0, which can be considered as artifacts. (**b**) At lower concentrations, the corresponding efficiency histogram exhibited a meaningful FRET population at <E> ≈ 0.55, that represented the apo-CaM state. The color code gives the number of burst for each point in the S vs. E plane.

**Table 1 biomolecules-12-00468-t001:** Comparison of different molar molecule concentrations as determined from *N* for different confocal detection volumes.

Pinhole Ø	30 μm	50 μm	75 μm	150 μm
	^1^ V ≈ 0.5 fL	V ≈ 1 fL	V ≈ 2 fL	V ≈ 4 fL
*N*	Molar Concentration in pM			
0.001	3.32	1.66	0.83	0.42
0.005	16.61	8.30	4.15	2.08
0.01	33.21	16.61	8.30	4.15
0.02	66.42	33.21	16.61	8.30
0.03	99.63	49.82	24.91	12.45
0.05	166.06	83.03	41.51	20.76
0.1	332.12	166.06	83.03	41.51

^1^ Effective detection volumes depend on the excitation and fluorescence emission wavelengths. The values given here represent a mean value, as obtained by averaging from 485 and 640 nm wavelength conditions.

## Data Availability

Data that are contained within the article are available on request from the corresponding authors.

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
