# Peer review of "Impact of Molecule Concentration, Diffusion Rates and Surface Passivation on Single-Molecule Fluorescence Studies in Solution"

_biomolecules, 2022, doi:10.3390/biom12030468_

Round 1

Reviewer 1 Report

A central challenge of solution-based single-molecule fluorescence experiments is their requirement of low (picomolar) concentrations to achieve single-molecule resolution. In their manuscript, Yukhnovets et al. address several key questions regarding multi-molecule events, the accurate quantification of molecular concentrations, and surface passivation.

The work is certainly of broader interest to users of single-molecule fluorescence and hence well suited for publication in biomolecules after revision. However, I would ask the authors to provide some additional data, explanations and clarifications that I believe will make the work more complete and accessible.

  1. The authors investigate the loss of fluorescence in solution during the measurement time. Would it also be possible to corroborate these results by investigating the deposition of molecules at the surface of the coverslide after the experiment?
  2. The estimation of the molecular concentrations, as described in section 2.10, is rather unconventional and relies on the assumption of equal diffusion time for all molecules. The concentration could also be estimated by FCS, which the authors use for the calibration of the confocal volume. The application of eq. 5 should be cross-checked with an FCS-based estimate of the concentration. Related, I am wondering Table 1, showing the straightforward relation between N and V is needed.
  3. The authors mention at different parts that the confocal volume for the different color channels deviate. What are the respective sizes for the experimental setup used?
  4. I am not following the equations for the chance coincidence event detection in section 2.11. Equation 7-9 seem to be given without any derivation/motivation. It also remains unclear how the parameters of the theoretical curves have been estimated for the comparison between measured and predicted chance coincidence events in Fig. 4. How where the burst durations in the two color channels estimated? From the equations, it is unclear how the threshold nbr is accounted for in the theoretical curves.
  5. The false-positive coincidence events should also be visible in a histogram of the apparent stoichiometry as shown in Fig. 5 for the smFRET experiments. It would be interesting to see the respective stoichiometry distribution for the BTCCD data shown in Fig. 1-2 to confirm the decrease of the fraction of acceptor-labeled molecules and to visualize the mixing between the donor and acceptor populations.
  6. The authors mention on line 333-335 that a count rate reduction was not observed at other GuHCl concentrations. This seems an important result and I would suggest that this data is shown.
  7. A minor point, the abbreviation used by the authors for guanidine hydrochloride seems unconventional. It should be GdnHCl (instead of GndHCl), however GuHCl or GdmHCl are more commonly used in my eyes.
  8. The equal propensity for sticking of the folded and unfolded states seems surprising given that the authors state that no sticking was observed for high and low GuHCl concentrations. What is the author’s hypothesis for this observation? I could imaging that the sticking is mediated by dye sticking (thus independent of the folded/unfolded state) which could be dependent on the change of the surface charge or hydrophobicity by the interaction of GuHCl with the coverslip.
  9. For the smFRET data shown in Fig. 5, why is the FRET population in b entirely missing in a? I understand that the mixing with donor or acceptor only molecules will contaminate the FRET species, however I would still expect a residual peak of the FRET population at the given concentrations. Likewise, how can a high-FRET species arise at high concentrations? This cannot be an effect of multi-molecule events. In the case of a donor only event, the stoichiometry would be shifted towards higher values and the FRET efficiency would be lowered due to the additional signal contribution in the donor channel. In the case of an acceptor only event, the stoichiometry would be shifted to lower values and no change of the FRET efficiency is expected because the coinciding molecules are most likely too far apart. What is the origin of the additional species at high FRET efficiency?

Author Response

Please find my reply in the attached file

Reviewer 2 Report

The aim of this study reported by Olessya et al., only one molecule in the detection volume at the time that is not only affected by the absolute molecule concentration, but also by the rate of diffusion. Therefore, they discuss approaches to control and to measure absolute molecule concentrations. Furthermore, they introduce an approach to calculate the probability of chance coincidence events and demonstrate that measurements with challenging smFRET samples require a strict limit of maximal sample concentrations in order to produce meaningful results.

Therefore, I recommend some Major revisions.  The major points of revisions are listed as follows;

  1. Authors must check typos, grammatical errors and punctuation. 
  2. Introduction must be improved
  3. Some significant relevant references from the last five years should be added in the introduction section.
  4. Improve the resolution of Figures
  5. Expand section for burst analysis; Two-color coincidence detection; chance coincidence probability

Author Response

Please find my reply in the attached file,

Reviewer 3 Report

This manuscript shows valuable findings that can be well-applied in single molecule FRET studies. The data is solid and well-presented, and the content is well-written. I suggest minor revisions.

Comments:

Figure 1 a&b panels are d bit difficult to read, and I don’t like the way the authors represent the data here. For F1a, I suggest to put the individual curves from the same channel (red or blue) at different time points together in one figure to show the trend better. For F1b, I suggest to make data of “Number of bursts” same color and that of “concentration” another color while make these two groups with different label shapes. And it can be applied to following figures containing data of AF488 and Atto647N.

Line 68, “TRIS” should be “Tris”.

Line 386, why don’t you use the same long wavelength dye for better comparison? Will the choice of dyes affect the results?

Line 419, have you tried other methods to remove excess free dye? What causes the low efficiency? What is the labeling rate for the double labeled RNC?

Author Response

(The authors gave the same response as above.)

Round 2

Reviewer 2 Report

The paper is accepted in the present form. 

Author Response

Thanks !